# Ωto_abR: A Web Application for the Visualization and Analysis of Click-Evoked Auditory Brainstem Responses

Aristotelis Ballas [1],* and Panagiotis Katrakazas [2],*

1   School of Electrical and Computer Engineering, National Technical University of Athens, 15773 Athens, Greece
2   Zelus P.C., 14452 Athens, Greece
*   Correspondence: telis.ballas@gmail.com (A.B.); p.katrakazas@zelus.gr (P.K.)

**Abstract:** Since its inception by Jewett and Williston in the late 1960s, the auditory brainstem response (ABR) has been an indispensable diagnostic tool, used by audiologists around the world. Click-evoked ABR testing proves to be a reliable tool, as it provides an objective representation of the auditory function, an estimate of hearing thresholds and the ability to pinpoint a potential issue in the auditory neural pathway. The present study describes state-of-the-art ABR analytics-related platforms and provides an overview of their functionality. In conjunction, we introduce the design and development of a newly developed, user-friendly web application, built in R language. This application provides several well-known and newly key characteristics for the analysis of ABR waveforms. These include absolute peak latencies, amplitudes, and interpeak latencies.

**Keywords:** click-evoked auditory brainstem response; ABR; ABR analysis platform; ABR web application

## 1. Introduction

Auditory brainstem responses (ABRs) are measures of electrical events generated within the auditory brainstem pathway [1]. ABR latencies are related to two aspects of brainstem changes, (a) loss of input from the cochlea and (b) changes in brainstem conduction velocity. Hearing-related problems are present when longer latencies exist for all ABR waves (namely waves I–V); however, additional problems can be found when checking the (a) absolute latency interaural difference of wave V, (b) the interpeak interval interaural difference among wave I–V, (c) the absolute latency of wave V, (d) the absolute latencies and interpeak interval latencies among waves I–III, I–V, III–V and (e) the absent auditory brainstem response in the involved ear [2]. This highlights that waves I, III, and V are the most common and most frequently used ABR indicators. Wave III and especially wave V, are the largest waves and are primary candidates for detecting hearing thresholds and identifying the remaining waves [3].

The last point, i.e., the identification of ABR waves and extraction of meaningful knowledge, fueled the current research as there has been a revamped interest in automated feature extraction [4] and time-series analysis [5] for revealing differences in the complexity between pathological and normal conditions quantitatively. Moreover, as also shown, there is a demand for developing an alternative research setup, driven by the need for "mobile, robust, inexpensive, flexible and modular extensible research platforms" [6]. The enhancement of graphical user interfaces suitable for the fast performance of test paradigms and robustness against errors while providing simultaneous calculation and minimizing errors by having an online tool, serves as an auxiliary means to enhance, and not compete, with commercial devices and overcome possible irritation caused by software conflicts in hospital systems, as shown by the increased satisfaction in similar cases of other healthcare areas [7–9].

Click-evoked ABR measurements serve as a reliable tool to cautiously estimate behavioral pure tone audiometry (PTA) thresholds at the average of frequencies of 1–4 and

2–4 kHz [10]. Studies over the last years have focused on waveform, latency and amplitude characteristics of the measurements [11] and comparison with other types of stimuli used [12,13] and threshold levels [14]. A normal ABR consists of up to seven positive wave peaks with subsequent valleys in between [15]. These positive peaks are typically labeled as Roman numerals (I–VII) and are referred to as Jewett peaks. Waves I through V are the primary components of the ABR. Wave I and II are originated in the VIIIth cranial nerve and occur at approximately 1.5 and 2.5 msec respectively, for click-evoked ABRs. As we move along the VIIIth nerve, wave III originates at the point of the cochlear nucleus and is typically recorded at 3.5 msec [16]. Finally, the last two waves are generated in a more rostral brainstem location. Wave IV is generated in the olivary complex and lateral lemniscus region and wave V in the lateral lemniscus and inferior colliculus region. They are most likely to be recorded around 4.5 and 5.5 msec, respectively [17]. Figure 1 is a visual representation of the neural generators mentioned earlier.

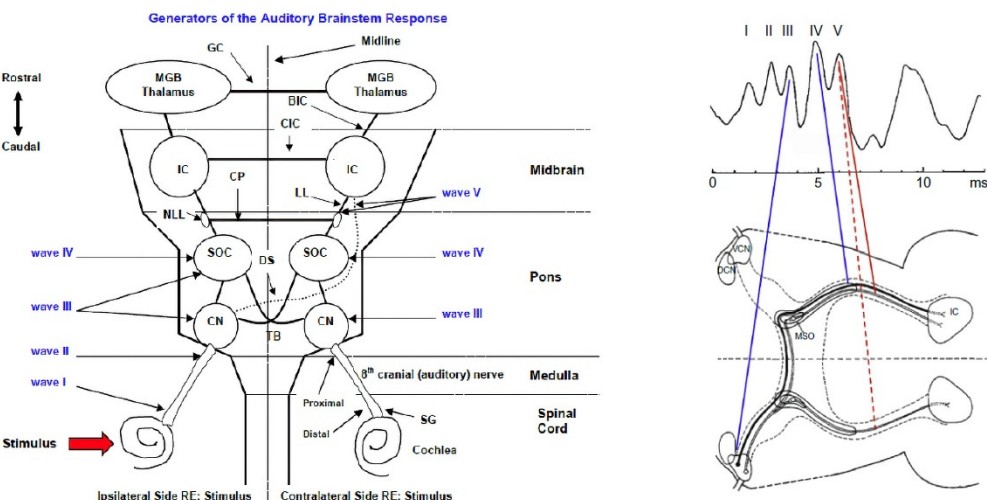

**Figure 1.** ABR neural generators. The left image was extracted from [17] and the right from [16]. Reprinted with permission from Ref. [17]. Copyright 2007 James Hall. Reprinted with permission from Ref. [16]. Copyright 2019 Jos Jan Eggermont.

To properly classify auditory brainstem responses (normal/abnormal), audiologists and clinicians emphasize the following characteristics of the waveform:

- Presence/absence of Jewett Peaks: the presence or absence of Jewett peas is an indicator of dysfunction in the auditory pathway;
- Peak amplitude (μV) reflects the number of neurons firing as a result of the stimuli;
- Peak latency (ms) reflects two aspects of brainstem changes, loss of input from the cochlea and changes in brainstem conduction velocity [2];
- Interpeak latency (ms) refers to the time difference between the onset of each wave;
- Latency/intensity function (LIF) is obtained by plotting the waves peak latency across the click intensity [18].

The main focus of this study is the development of an automated tool which will provide audiologists and clinicians the means to further analyze ABR waveforms. To this end, we emphasized on the platform's analytic capabilities and used an ABR dataset to demonstrate said capabilities. The above points are described in detail in the following sections.

## 2. State-of-the-Art in ABR Analytic Platforms

Prior existing tools for the ABR analysis consist of commercially available devices used for such purposes. The Interacoustics Eclipse platform (module EP15) [19], the BIOPAC MP system [20], the Intelligent Hearing system SmartEP [21], the Neurosoft OAE screening system NEURO-AUDIO [22], and the Brain Products EP-PreAmp [23] are among the most commonly known platforms offering capture, monitoring and interpretation of ABR signals

within their proprietary software. Moreover, individual attempts (e.g., sABR software [24] from the University of British Columbia and a Telemedical System for the Evaluation of ABR's [25] by the AGH University of Science and Technology) are not fully exploiting the potential of the ABR analysis. However, a web and publicly available solution are not offered, resulting in the creation of data lakes established "on premises" within a hospital's or clinical center's data center, if it exists. This creates a huge loss in knowledge creation and a missing opportunity for identifying emerging patterns or important information that can be extracted from the ABR signals.

With the development of the presented application, we intend to provide an automated way for clinicians and physicians to further exploit the ABR measurements as an online audiological tool, identifying patterns and extracting features out of them, towards the creation of an enhanced knowledge base. The main aim of this research is the development of a web tool for the import, characterization and classification of click-evoked ABRs, along with the proposal of an ABR metric; the ABR area under curve (AUC).

## 3. Materials and Methods

Prior to developing a platform, understanding the information embedded in existing ABR datafiles was essential. Data from the Interacoustics Eclipse Platform (module EP15) were used for this purpose. The Eclipse EP15 module offers the option to export the raw data measurements of a test session and save them as an extensible markup language (.xml) file. While a public XML Schema for the exported files is not available, the XML header description can be found in the manufacturer's "Additional Information" manual (a copy of the "Additional Information" manual can be found at: https://www.manualslib.com/manual/1365514/Interacoustics-Eclipse.html?page=1#manual (accessed on 20 September 2021)) for the EP15 module. The stimulus rate of each session was 11 stimuli per second and the waveform intensity varied from 70 dB up to 100 dB (with a step of 10 dB), above normal hearing level. From the data used in the current study, "normative" data from a total of 146 patients were obtained, thus creating the LIF for waves I, III and V.

The R language (https://www.r-project.org/about.html (accessed on 20 September 2021)) and its related libraries (e.g., XML, Bolstad2, Quantmod) were used to parse the exported .xml files, extract the related data measurements, reconstruct, visualize and finally, analyze the ABRs at a later stage.

The steps and procedures followed in reconstructing and analyzing each patient's ABR test are listed below, along with a short description for better understanding:

1. Selection of .xml file: user selects the .xml file of a patient's test;
2. Parsing of .xml: the selected .xml file is parsed by R and all information is saved into a dataframe;
3. Extract raw data of ABR test: after parsing the XML file, measurements associated with the amplitude of the potential are extracted and saved into a second dataframe;
4. Convert raw data to μV: the extracted amplitude data are converted into μV;
5. Detection of ABR waveform's peaks and valleys: potential Jewett peaks and valleys of the ABR are automatically detected and presented to the user;
6. Visualization of ABR waveform: the ABR waveform and potential peaks and valleys are plotted;
7. User input: users are able to add, remove or correct the proposed Jewett peaks and valleys;
8. Analyze waveform characteristics: Absolute peak latency and inter-peak latency are calculated. The areas are calculated;
9. Visualize Analysis: all Latency/Intensity functions are plotted.

As the first five Jewett peaks are typically present up to 5.5–6 ms of the ABR, we based our analysis up to 7.5 ms the waveform. However, in our platform, the full length of the waveform (i.e., 15 ms), which is the exact length of the Eclipse EP15-extracted .xml file is also displayed, as a user might require a full waveform examination for a thorough analysis.

As already mentioned, the analysis of an ABR waveform is inextricably linked with the detection of the present Jewett peaks and valleys (step 5). To that end, we emphasized

our work on the correct detection of the waveform's key characteristics and developed a procedure accordingly. The procedure begins by detecting all the extrema in a patient's ABR potential measurements. The calculated maxima are marked as potential Jewett peaks and the minima on each side of the maxima, as valley points. However, it is possible that more than five maxima are detected in a patient's waveform. The flow diagram in Figure 2 describes the logic behind the final selection of the Jewett peaks.

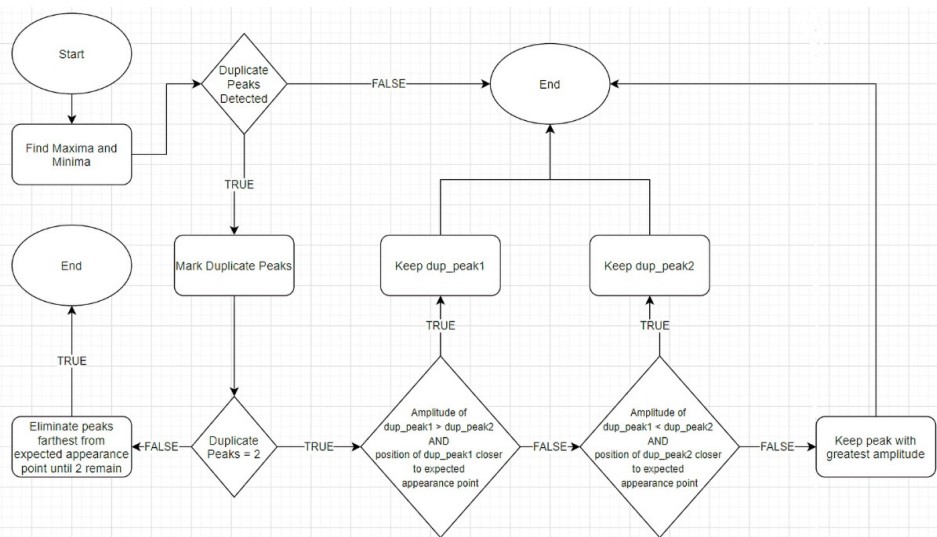

**Figure 2.** Flowchart of Jewett peak selection process.

## 4. Results

To parse, extract, and analyze the needed data from the Eclipse's .xml files, several functions in the R language were developed. After successful testing of the functions, a new R package was built and named "abr". The next step was to develop an interactive and dynamic, user-friendly web application, using R's "Shiny" package (https://shiny.rstudio.com/ (accessed on 20 May 2021)). A Shiny app, such as Ωto_abR, consists of two objects: the user interface (UI) side and the server side. The UI object controls the layout and appearance of the application (hereinafter mentioned as app), and the server contains all the R code that is needed to bootstrap the two objects. In this project, we decided to bundle the code for both the UI and server side, in a single R file, named app.R. In order to deploy the app on the web cloud services of shinyapps.io (https://www.shinyapps.io/, accessed on 7 July 2021), our "abr" package had to be uploaded on a repository, such as GitHub (https://github.com/ (accessed on 22 September 2020)). A visual representation of the app's architecture is presented in Figure 3.

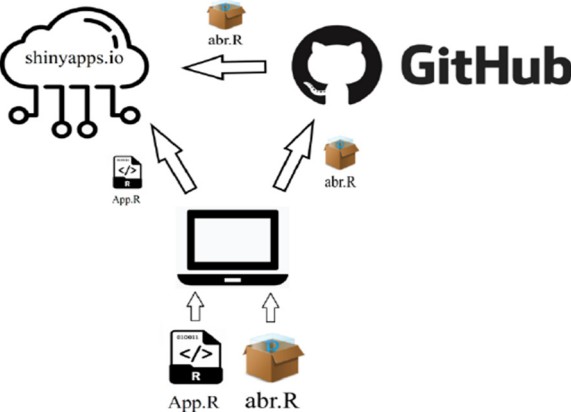

**Figure 3.** Architecture of the Ωto_abR web application.

### 4.1. Home Page

The "Home" page of the app is shown in Figure 4. On this page, the app "waits" for the user to upload a patient's .xml file. Upon upload and under the "ABR Plots" subtab, the ABR waveform and the automatically detected Jewett peaks and valleys are plotted, as shown in Figure 5. In addition, under the "ABR Plot-Extended" subtab, the full ABR (up to 15 ms) is displayed (Figure 6). Furthermore, users are given the option to manually change the automatically detected peaks and valleys on the waveform. By alternating these values, the waveform's analysis is, in turn, dynamically converted to match the aforementioned values. The above options are depicted in Figures 7–10.

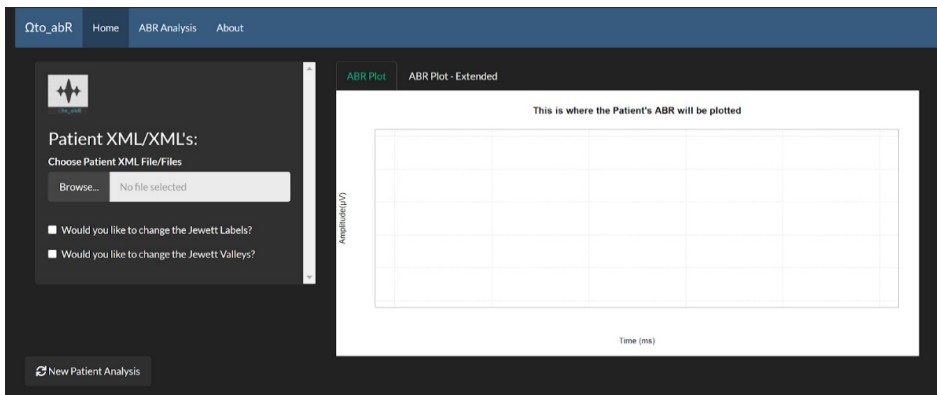

**Figure 4.** Home page.

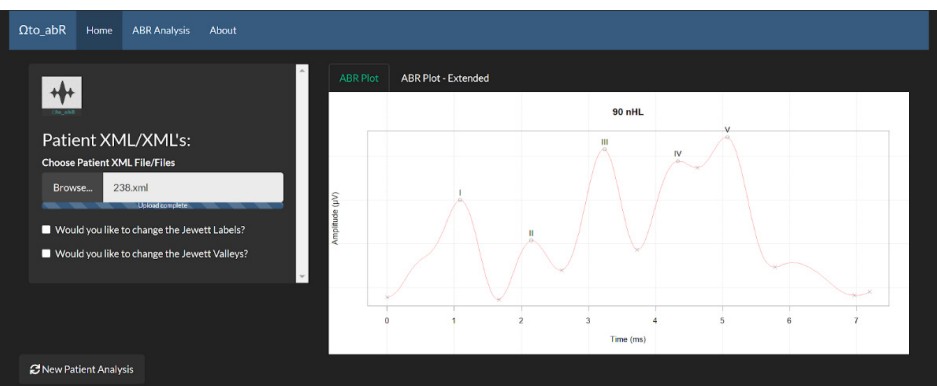

**Figure 5.** ABR plot.

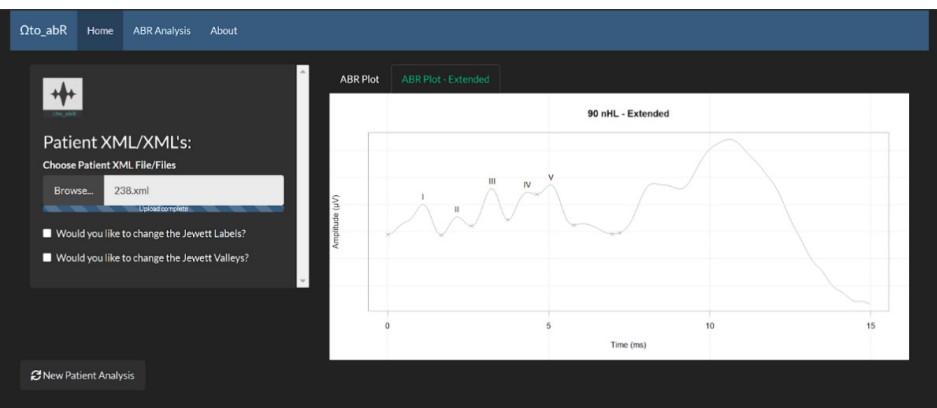

**Figure 6.** ABR Plot–Extended.

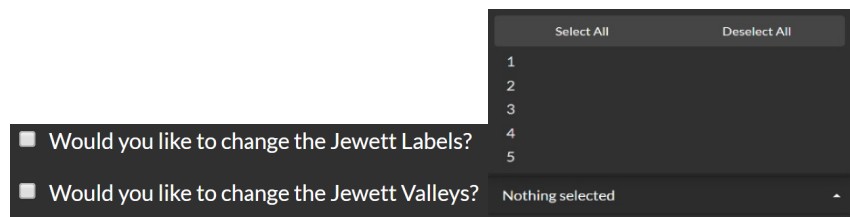

**Figure 7.** Select whether and which peaks or valleys should change.

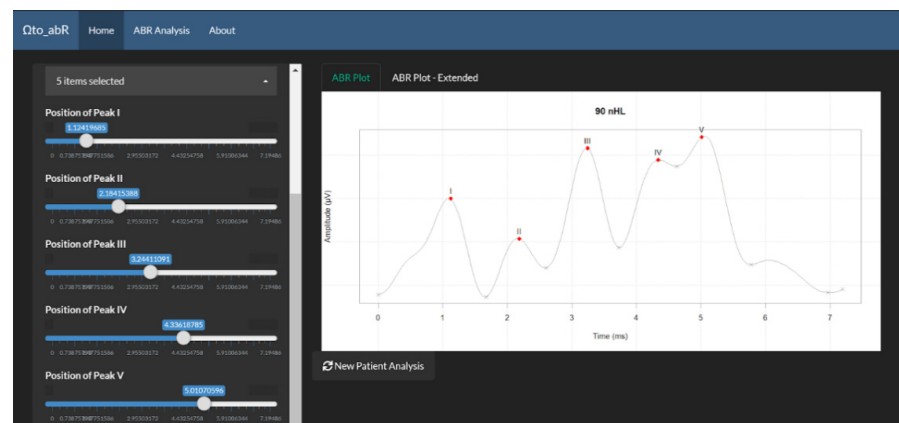

**Figure 8.** Slider controls and changed peaks.

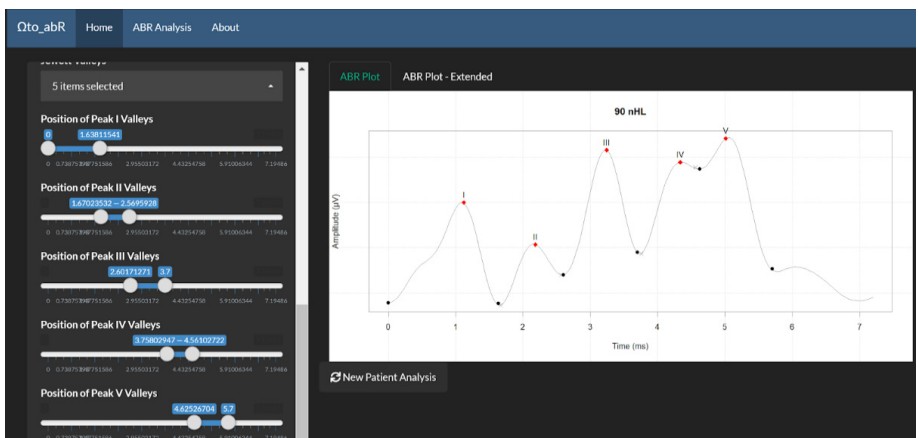

**Figure 9.** Slider controls and changed valleys.

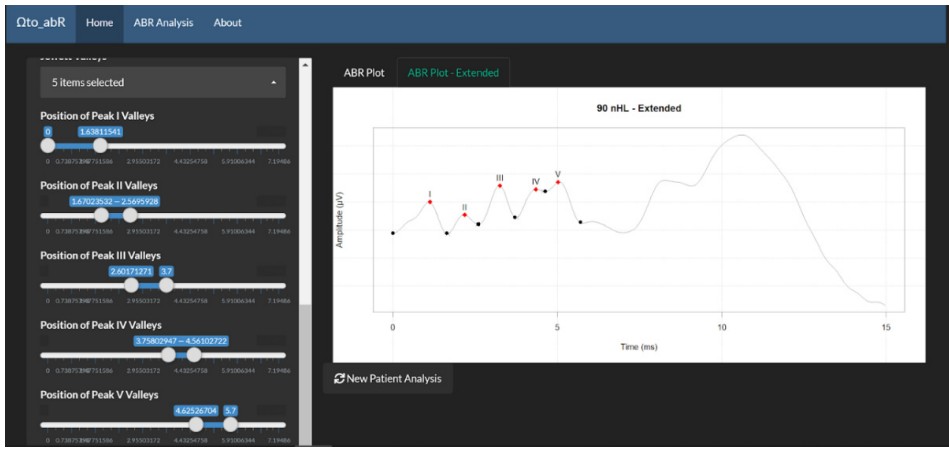

**Figure 10.** ABR extended-changed values.

### 4.2. ABR Analysis Page

The ABR Analysis page, displays visual representations of the ABR waveform analysis (Figure 11). There are four analysis plots in total and there is a subtab for each one. The mentioned plots, shown in Figure 12, consist of a plot for the ABR area analysis and plots depicting the latency/intensity functions for peaks I, III and V. The ABR area under curve (AUC), is a suggested and newly introduced metric, by our team. It is measured in $\mu V^2$, and Simpson's rule [26] is used for its estimation. Simpson's rule is used for numerical approximation of definite integrals. The general form of the rule is described by the following equations:

$$\int_a^b f(x)dx \approx \frac{\Delta x}{3}(f(x_0) + 4f(x_1) + 2f(x_2) + 4f(x_3) + \ldots + 2f(x_4) + \ldots + 4f(x_{n-1}) + 4f(x_n)) \tag{1}$$

where

$$\Delta x = \frac{b-a}{n} \tag{2}$$

and

$$x = \alpha + i\Delta x. \tag{3}$$

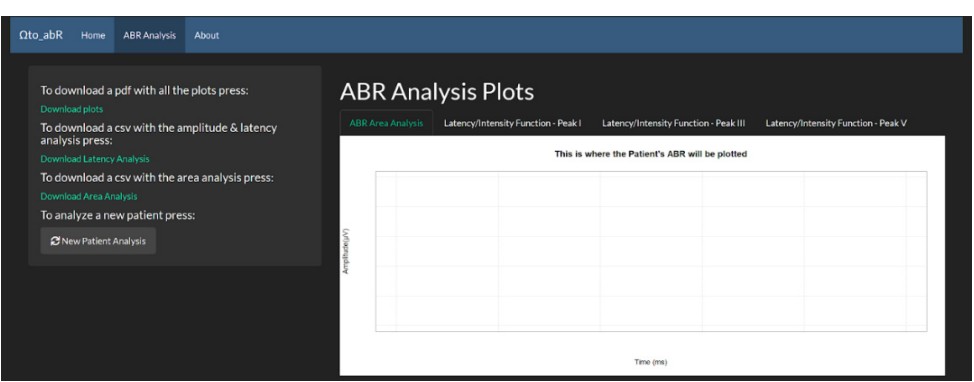

**Figure 11.** ABR Analysis tab.

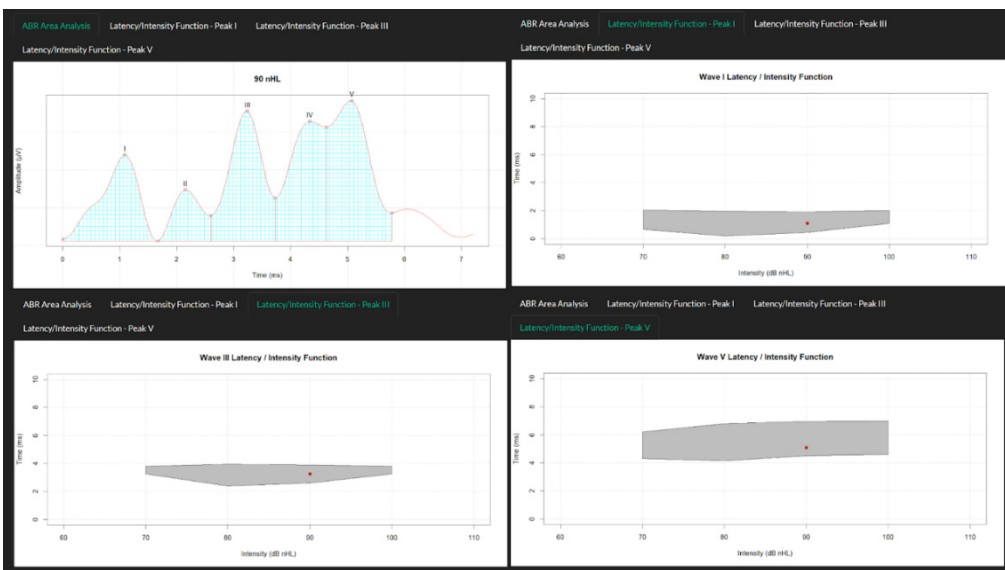

**Figure 12.** ABR analysis plots.

Indicative measurements of the AUC for the waveform in Figure 8 are presented in Table 1. The AUC measurement is plotted and displayed in the platform, as shown in Figure 13. The idea behind this approach is to further investigate features and characteristics that might lie within the area of the drawn plot. To the extent of our knowledge, this

has been investigated in a past bibliography and might provide some additional insight to ABR measurements.

**Table 1.** Indicative measurements for AUC metric.

| Curve | I | II | III | IV | V |
|---|---|---|---|---|---|
| **AUC** ($\mu V^2$) | 1.03 | 0.56 | 0.81 | 0.68 | 0.89 |

**Figure 13.** Area under curve (AUC) measurement through Simpson's rule.

In the same page, users are provided the option to save the plots in a .pdf file and the amplitude, latency, and area analysis to .csv files. Finally, a "New Patient Analysis" button is also available.

## 5. Discussion

The $\Omega$to_abR platform aims to be an easily accessible and user-friendly web application to aid clinicians and physicians in the diagnosis and analysis of the click-evoked ABR test. As an audiological tool, the app offers capabilities of automatic detection, extraction, and exploitation of key characteristics of the waveform (e.g., detection of Jewett peaks I-V, peak latency and interpeak latency). As previously mentioned, if needed, users can change or modify the Jewett markers on the ABR waveform and as a result, the analytics metrics are adjusted dynamically. Moreover, the added value of our approach lies in the fact that clinicians and physicians are provided the ability to remotely examine previous ABR data of a patient (e.g., from their home) without needing to be physically present in the audiological equipment room, simply by uploading the appropriate files. They will be able to ascertain a diagnosis and classify the ABR in question or assist a fellow physician by offering remote consultation.

By using the R language and by extension the Shiny package, the platform is developed in a way that allows the addition of new features and capabilities in the future (e.g., overlap of two or more different ABR waveform analyses in a single plot). Future work also involves creating a publicly available database which will consist of data extracted from the proposed platform. As a first step, we intend to deploy the web-app to a standalone server, under an institutional domain, which will not be bound to shinyapps.io. As a current ongoing work, data acquisition from already contacted clinics is in place, so the extracted data can be subject to additional analysis in order to identify data patterns and relationships. The newly proposed metric (AUC) will be also further analyzed and studied in an upcoming publication, given the timely acquisition and analysis of the aforementioned clinical data. Finally, identification and extraction of knowledge from other platform ABR recorded files will be pursued to extend the existing functionalities of the platform. The goal is to create a file-agnostic ABR-analysis tool to provide easily comprehensible analytic functionalities to clinicians and doctors of the audiological field, overcoming any limitations they may encounter by commercial devices and proprietary software.

**Author Contributions:** Conceptualization, A.B. and P.K.; validation, A.B. and P.K.; formal analysis, A.B.; investigation, A.B.; resources, P.K.; data curation, A.B.; writing—original draft preparation, A.B.; writing—review and editing, P.K.; visualization, A.B.; supervision, P.K.; project administration, P.K. All authors have read and agreed to the published version of the manuscript.

**Funding:** This research received no external funding.

**Institutional Review Board Statement:** Not applicable.

**Informed Consent Statement:** Not applicable.

**Data Availability Statement:** The "Ωto_abR" platform is available online at https://autoabr.shinyapps.io/_abr_shiny_final/ (accessed on 7 July 2021). The code for the UI part of the platform, along with sample files for the use of the platform are available online at https://github.com/aballas22/Oto_abR (accessed on 7 July 2021). The rest of the anonymized .xml files, which were used for the presented analysis, were provided from a private clinic and are not licensed for public use.

**Acknowledgments:** The authors wish to thank Interacoustics A/S and specifically their technical support team for providing additional insight on the Eclipse software.

**Conflicts of Interest:** The authors declare no conflict of interest.

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
