# Peer review of "Ωto_abR: A Web Application for the Visualization and Analysis of Click-Evoked Auditory Brainstem Responses"

_digital, doi:10.3390/digital1040014_

Round 1
Reviewer 1 Report
I have been reading your manuscript, and I have several questions related to it.
About table 1, I'm wondering whether the dataset used to estimate those values is available, as well as its metrics, like number of patients, distribution of age and gender, diagnosis of some impairment, ...
I have been having a look at the XML files you have provided in the GitHub repository https://github.com/aballas22/ABR-Sample-FIle , and I have realized from their headers there should be some XML Schema in some place. I'm wondering whether the device manufacturer, creator of the export tool you have used to obtain the XML files, documented this format in some public place.
A more generic question related to the documentation of this XML file format is whether there is some agreed, standardized representation for this kind of data which is "device manufacturer agnostic".
I also have missed some documentation of the sample dataset: origin, distribution licencing, whether it is anonymized, etc...
When I tried accessing the platform following link at lines 231-232 https://www.shinyapps.io/admin/#/application/2452229 , and it did not work. When I tried, ShinyApps requested me to log in in order to access, and after that it told "Not found". Could you provide a public link to your deployed platform, please?
I tried finding the source code of your developed Shiny app, and I could not find it. Could you publish it in some public repository, with the proper distribution licencing, please?
Formulas from 161 to 167 should appear in higher quality, as they appear a bit blurry.
Reviewer 2 Report
This paper describes a new web-application developed with the R language. The web application allows for visualization and analysis of several ABR characteristics including some newly introduced here. This aids clinicians and physicians immensely. In addition, the paper overviews, and reviews existing applications and which are compared to the proposed web-application. I think the paper reads well and should be published. I would recommend the authors proofread the paper carefully and consider re-wording sections that read as “the authors intend to …” or “it was the authors aim to …”, to a more direct style e.g. “we intend to …” and “we aimed to …” Title – Consider renaming the application as greek symbols/characters might not be easy to type for many people, and possibly could make searching/finding the application harder. Abstract – Consider some rewording on the sentences starting from line 12, “The present paper (…)” and 13, “In conjunction, (…)”. Line 14, consider rewording to “user-friendly Shiny web application, build with the R language ”? Line 15-16, consider splitting the line as follows: “(…) which provides several well-known and newly developed key characteristics for the analysis of ABR waveforms. These include absolute peak latencies, amplitudes and interpeak latencies. Introduction Consider rewording sentence starting from line 33. Sentence starting from line 39 is confusing and should be split into two or even three, so that the points come across more easily. The paper is brief enough that I think we would not need to have a description of the contents of each section as laid out at the end of introduction. More useful would be a wrap up of what has been done and achieved in this paper, similar what is described in the end of the abstract. Section 2 Line 89, consider rephrasing as e.g “In this work we wanted to provide (…)” Section 3 Line 131, consider rephrasing around “said characteristics”. This section is missing some description of how the web-application is put together with Shiny. Without going into too much detail, maybe it could also describe how it is hosted. Section 4 Equations should be typeset again, as these seem a bit pixelated. Section 5 Would be good to get open source from day 1, so consider opening up development on github or other. Additionally, hosting the shiny application under an institutional domain would be much preferred.Author Response
Please see the attachment

Round 2
Reviewer 1 Report
First of all, I recommend you to treat URLs embedded in the manuscript carefully in your word processor, as right now they are treated as text, and it can happen like in line 240, where the URL is split between two lines, adding automatically the word processor a hyphen. Readers who are transcribing the URL "in the old way" are going to put also the hyphen, happening they are not going to reach the web site.
I might be being a bit picky here, but I think there is a sentence which should be lightly reworded at lines 242 and 243. You were explaining in line 241 there available xml sample files are available in the GitHub repo, and just in lines 242 and 243 you are talking about that anonymized xml files used for the analysis are not available. Someone reading all these sentences in a sloppy way could misunderstand what you are meaning there.
Line 97: I guess that in the sentence "was" should be "were"
I have been using https://autoabr.shinyapps.io/_abr_shiny_final/ , and it does what it was described at the manuscript. As I can see there lots of potential after I have been playing with it and a couple of input files, I have a couple of suggestions for future developments. It could be interesting to overlap two or more different ABR waveform analyses in a single plot, either normalizing or synchronizing some or all the peaks or valleys in the input data in some way.
I have also been having a look at the updates in repo https://github.com/aballas22/Oto_abR . A recommendation is adding some README.md or similar with a minimal documentation explaining what it is in the repo.
I was having a look at the Shiny R code you uploaded at the repo, weighing its viability. I realized you commented out a couple of libraries called `abr2` and `autoabr`. As I was curious I searched for "abr" keyword in the whole file, and I realized that code would not work "as is" because it depends on function `abr_test`:
https://github.com/aballas22/Oto_abR/blob/b4c24a03b363f41c695ecf3882cca250ebe14b26/app.R#L389
which is not declared there. I searched for it in internet, and I found it seems to be defined inside `autoabr` library, based on the findings in this online documentation https://rdrr.io/github/aballas22/autoabr/api/ .
In that side I found next page https://rdrr.io/github/aballas22/autoabr/ tells that the repo of the package is available at https://github.com/aballas22/autoabr/ , but the repo cannot be found there (is it a private repo?). So, either the `autoabr` R library should also be publicly available, or that call to `abr_test` should be substituted in some way.
